# Tree Ring Width Responses of *Pinus densiflora* and *Robinia pseudoacacia* to Climate Variation in the Mount Tai Area of Northern China

Yuan He [1,†], Qinghui Yu [1,†], Guifang Wang [1], Ming Hao [1], Simin Fan [1], Dingmeng Hu [2,\*], Zongtai Li [2] and Peng Gao [1,\*]

1   Mountain Tai Forest Ecosystem Research Station of State Forestry and Grassland Administration, Forestry College, Shandong Agricultural University, Tai'an 271018, China
2   Shandong Academy of Forestry, Ji'nan 250014, China
\*   Correspondence: gaopengy@163.com (P.G.); hudingmeng@shangdong.cn (D.H.)
†   These authors contributed equally to this work.

**Abstract:** To effectively combat climate change and put plans in place to enhance the health and quality of forests, research on the connections between long-term climate change and tree ring width is essential. Here, *Pinus densiflora* Siebold & Zucc. and *Robinia pseudoacacia* L. in the Mount Tai area were studied. Specifically, their tree ring width characteristics were investigated according to the principles of dendrochronology, based on the analysis of multiyear climate data (1972–2022). The results showed that (1) the variation in tree ring width of *Pinus densiflora* Siebold & Zucc. generally decreased. Its basal area increment sequence presented a "growth-decline" change process. The trend for *Robinia pseudoacacia* L. was stability at first and then a reduction. Its basal area increment sequence presented a "growth-stabilization" change process. (2) The standard chronologies of both tree species contained more environmental information than the residual chronologies, rendering the former more appropriate for analysis. (3) The tree ring width of *Pinus densiflora* Siebold & Zucc. was positively correlated with the mean maximum temperature in February, March, and July, the mean temperature in May, and the mean precipitation in December of the previous year and June of the current year. The tree ring width of *Robinia pseudoacacia* L. was positively correlated with the mean and maximum temperatures in October of the previous year, the mean maximum temperature in May of the current year, and the mean temperature in July of the current year. This variable was positively correlated with the mean precipitation in November of the previous year and July of the current year but negatively correlated with the mean minimum temperature in November of the previous year and the mean temperature in December of the current year. (4) During 1987–1988, 1994–1995, 2005–2006, and 2018–2019, the tree ring width was restricted by rising temperatures and low precipitation, with variations in tree ring width observed in various hydrological periods. (5) The moisture index, followed by the warmth index, precipitation and temperature, was the primary climate factor determining the tree ring width in both tree species.

**Keywords:** tree ring width; climate change; *Pinus densiflora* Siebold & Zucc.; *Robinia pseudoacacia* L.; Mount Tai

## 1. Introduction

In recent years, extreme weather frequency and intensity have increased [1–3]. The impacts of global climate change, characterized by warming and drought, are not limited to humans but extend to the productivity and functional stability of forest ecosystems [4,5]. This phenomenon has posed a major threat to forest productivity, and has even caused forest decline and death, ultimately resulting in irreversible damage to the forest ecosystem [6–10]. Due to their reliable dating, high continuity, widespread geographic distribution, and high temporal precision (annual or seasonal), tree rings have emerged as a significant source

of information on historical climate and environmental change [11–13]. Investigating the correlations between climate factors and tree ring width holds significant value in comprehending the response mechanisms of forest ecosystems to climate change within a specific region.

The forest ecosystem is a type of terrestrial ecosystem, and its productivity accounts for 50% of the terrestrial ecosystem [14,15]. Trees are a major component of the forest ecosystem, and their trunk growth, as indicated by ring width, is the main mode of biomass accumulation [16–18]. The investigation of tree ring width changes can provide insights into the growth status of trees over the years and help predict the impact of future climate change on tree growth [19,20]. Tree ring width, serving as a critical indicator for assessing tree growth, provides a valuable foundation for understanding the long-term fluctuations in tree ring width and growth patterns [21,22]. Studies have shown that tree ring width is influenced by both environmental and genetic factors of the tree itself [22,23]. Under the comprehensive influence of various factors, tree ring width displays a distinct trend, with climate factors being the primary elements that impact the physiological processes of trees [24,25]. Thus, building a model of the relationships between tree ring width and climate factors is beneficial for understanding the environment in which trees exist and their response mechanisms to various climate factors.

Numerous investigations have shown that the effects of climate factors on tree growth present temporal and spatial variations [26]. Due to geographic location and tree species, the impact of climate factors on tree ring width varies [27,28]. Compared to broad-leaved species, coniferous species exhibit clearer tree ring boundaries and fewer missing and false rings and thus yield more precise cross-dating results, making them an ideal choice for tree species in sampling and analysis [29,30]. Salzer, et al. [31] demonstrated that the chronology of near-treeline *Pinus longaeva* was effective in revealing changes in tree ring width and climate response. Olivar, et al. [32] reported that different types of conifers growing in the same environment had different tree ring widths in response to climate factors. Huang, et al. [33] established a growth climate database of *Pinus massoniana* Lamb. based on tree ring width data. They found that even though temperature and precipitation had equal effects on *Pinus massoniana* Lamb. growth in different areas, high summer temperatures remained the primary limiting factor. Broad-leaved species exhibit substantial differences in woody properties compared to those of coniferous species [23,34]. Despite the challenges posed by the hard wood and complex rings of broad-leaved species, tree ring width studies in these species remain valuable. This is because both species experience the same or similar climatic conditions, and it is climatic influence that is different. Takahashi et al. [35] studied the response of three broad-leaved tree species in eastern Canada to climate warming. The findings indicated that the impacts of global warming, such as drought stress, have endangered the growth of forests in the area. Shen et al. [36] studied the response of tree ring radial growth of three tree species (*Betula ermanii*, *Fagus crenata* and *Quercus crispula*) in cold temperate forests in central Japan to climate, and found that the tree ring width of the three tree species was negatively correlated with the average and minimum temperature of the previous autumn.

Mount Tai is situated in the central region of Shandong Province, China. This region is an exemplary mountainous area in northern China and has a warm, semi-humid continental climate. The region boasts significant forest cover, characterized by highly diverse and complex forest types and high productivity. Additionally, it is recognized for its species diversity and contribution to the global carbon sink, and plays a crucial role in preserving the ecological security of northern China and the wider East Asian region [37,38]. *Pinus densiflora* Siebold & Zucc. (PD) and *Robinia pseudoacacia* L. (RP) are recognized as critical vegetation and group species in this area. They play a significant role in restoring regional vegetation, controlling soil erosion, and enhancing the climate and environment of the region. To date, research on PD and RP in this area has mainly focused on soil improvement, soil hydrological effects, and the spatial heterogeneity of carbon, nitrogen, and phosphorus [39–41]. There have been few reports on the characterization of

tree ring width in PD and RP [42]. This phenomenon poses a major challenge in accurately assessing and scientifically addressing forest ecological afforestation projects and their related ecological service functions.

The aim of this study is to shed light on how forest tree ring width reacts to long-term climate changes in the region, with a focus on studying PD and RP in the Mount Tai area, a typical region characterized by mountainous terrain and soil conditions in northern China. The study was conducted with dendrochronological theory as a foundation by establishing tree ring chronologies for both PD and RP using climate data from 1972 to 2022. The main objectives were (1) to study the growth rate and change characteristics of PD and RP; (2) to determine the basic parameter values of the tree ring chronology and discuss the quality of the tree ring chronologies of PD and RP; (3) to investigate how the climate factors of a single month affect the tree ring width of PD and RP; (4) to investigate the pattern and characteristics of trends in PD and RP tree ring width due to multiyear cumulative climate factor changes; and (5) to examine the mechanism and process by which changing climate factors affect the tree ring width of PD and RP, as well as the links between these widths and numerous climate factors.

## 2. Materials and Methods

### 2.1. Site Description

This research was conducted at the Taishan Forest Ecosystem National Positional Observation and Research Station in the Huangqian watershed, Tai'an city, Shandong Province. The study area has geographic coordinates of E117°04′–117°22′ and N36°16′–36°28′ and a total watershed area of 292.20 km$^2$ (Appendix A Figure A1). The Huangqian River is a tributary of the Huanghe River, and the Huangqian River Basin is located in the national key area of soil erosion control in Mount Tai. The soil type in the study area is Haplic Luvisol, formed by granite gneiss, as classified by the World Reference Base for Soil Resources (WRB). With a mean thickness of 30 to 40 cm, the soil layer in the study area is characterized by a slightly acidic pH value ranging from 5.5 to 6.5. The average gravel concentration is 65.96%, while the soil moisture level ranges from 19.35 to 23.73% [41]. The climate is a warm, temperate, continental, semi-humid monsoon climate with a frost-free period of 196 days. The vegetation in the study area comprises warm, temperate, deciduous, broad-leaved forest and coniferous forest, which are home to various plant species, such as *Robinia pseudoacacia* L., *Pinus densiflora* Sieb. et Zucc., *Quercus acutissima* Carruth., and *Larix kaempferi* (Lamb.) Carr.

### 2.2. Sample Collection and Tree Ring Chronology Construction

Three typical sample plots (100 m × 100 m) in each of the *Pinus densiflora* and *Robinia pseudoacacia* forest areas were selected as study areas for tree core sample collection (a total of 6 plots) (Table 1). Eight trees with good growth and no trunk damage or decline in tree vitality were selected as sample trees in each plot. Sampling followed the International Tree-Ring Data Bank (ITRDB) standards. To ensure accurate cross-dating results, the tree core was drilled from two different directions (parallel and perpendicular to the slope) at chest height (1.3 m above the ground). Two tree cores were drilled into each tree. Twenty-four sample trees were selected in each strand (48 in total for both strands). In this experiment, a total of 96 sample tree cores were obtained. The collected core samples were stored in transparent straws labeled with sample information, sealed, and marked with a pen. The pretreatment steps included air-drying, fixing, and polishing the samples, after which cross-dating was performed under a microscope. The LINTAB5.0 tree ring analyzer, which offers a high degree of precision and accuracy with a measurement accuracy of 0.01 mm, was used to measure the tree ring width. The COFECHA program was used to cross-check tree ring data to ensure precise data and measurements [43]. Standard (STD) and residual (RES) chronologies were produced for the tree ring width data using the ARSTAN tool (Table 2) [44]. The original sequence of tree ring width was de-trended by using the negative exponential function of the ARSTAN program and the spline function whose step



size was 2/3 of the sequence. The STD was established by the double-weight averaging method [45,46]. At the same time, basal area increment (BAI) calculations were performed. The formula for the basal area increment is as follows: BAI $= \pi \left( r_n^2 - r_{n-1}^2 \right)$, where $r_n$ and $r_{n-1}$ are the radii of trees at breast height in years $n$ and $n-1$.

**Table 1.** Information on the basic growth indicators of *Pinus densiflora* Siebold & Zucc. and *Robinia pseudoacacia* L. in the study area.

| Species | Sampling Point | Longitude and Latitude | Altitude(m) | Slop (°) | Slope Direction | Tree Height (m) | Canopy Density | Density (Tree·hm$^{-2}$) |
|---|---|---|---|---|---|---|---|---|
| PD | 1 | N36°20′22″ E117°6′45″ | 800.0 | | Northwest | 9.28 ± 0.37 | 0.60 | 621.0 |
| | 2 | N36°20′22″ E117°6′40″ | 802.0 | 21.6 | | 9.19 ± 0.17 | | |
| | 3 | N36°20′22″ E117°6′49″ | 784.0 | | | 9.20 ± 0.27 | | |
| RP | 1 | N36°19′51″ E117°6′27″ | 760.8 | | Southeast | 11.20 ± 0.27 | 0.70 | 783.5 |
| | 2 | N36°20′23″ E117°5′41″ | 768.1 | 24.1 | | 10.93 ± 0.11 | | |
| | 3 | N36°19′87″ E117°5′45″ | 779.0 | | | 11.41 ± 0.31 | | |

**Table 2.** Statistics of tree ring width and tree ring chronology of *Pinus densiflora* Siebold & Zucc. and *Robinia pseudoacacia* L.

| Species | PD | | RP | |
|---|---|---|---|---|
| **Statistics** | **STD** | **RES** | **STD** | **RES** |
| Mean | 1.02 | 1.69 | 0.99 | 0.68 |
| Median | 0.98 | 1.68 | 1.00 | 0.69 |
| Standard deviation | 0.16 | 0.26 | 0.08 | 0.05 |
| Mean Sensitivity | 0.211 | 0.193 | 0.168 | 0.156 |
| Signal-to-noise Ratio | 40.931 | 38.557 | 28.511 | 27.369 |
| First-order Autocorrelation Coefficients | 0.112 | 0.114 | 0.126 | 0.154 |
| Mean Correlation Coefficient Between Trees | 0.529 | 0.518 | 0.522 | 0.513 |
| Expressed population signal | 0.977 | 0.975 | 0.934 | 0.908 |
| The variance of the First Principal Components | 47.91% | 45.43% | 50.07% | 48.19% |

STD is the standardized chronology and RES is the residual chronology.

### 2.3. Weather Data Sources

By selecting the meteorological data of the Tai'an Meteorological Bureau from 1972 to 2022 (http://sd.cma.gov.cn/gslb/tasqxj/, accessed on 1 January 2021), the time scale of the climate data was ensured to meet the research standards of dendrochronology and climatology. The data were sorted into monthly (mean maximum temperature per month, $T_{max}$; mean minimum temperature per month, $T_{min}$; mean temperature per month, $T_{mean}$; mean precipitation per month, $P_{re}$) and yearly (mean yearly temperature, $T_i$; mean yearly maximum temperature; $T_h$; mean yearly minimum temperature, $T_l$; mean yearly precipitation, $P_i$; moisture index, $M_i$; warmth index, $W_i$) values. The formula for the warmth index is as follows: $W_i = \sum_{i=1}^{n} (t_i - 5)$, where $n$ is the number of months with an average temperature over 5 °C, $t_i$ is the average temperature over 5 °C [47]. The formula for the moisture index is as follows: $M_i = \frac{P_i}{T_i}$. Meteorological data from September of the previous year to December of the current year were chosen for examination due to the "lag effect" of climate factors on tree ring width [48].

Between June and September, the study area has the most precipitation, accounting for 75.69% of the yearly precipitation (Figure 1A). Between 1972 and 2022, the study area temperatures generally increased (Figure 1B). By utilizing the P-III frequency curve method

to analyze the rainfall data over multiple years, it was determined that $C_s = 3.0C_v$ ($Cs$ represents the runoff coefficient of the watershed, and $Cv$ represents the coefficient of variation of the runoff). The results obtained by cross-reference to the data table indicated that a precipitation frequency of 75% corresponds to a precipitation level of 813.49 mm, while a precipitation frequency of 20% corresponds to a precipitation level of 1282.81 mm in the study area (Appendix A Table A1) [49,50]. Based on these values, wet years were identified as those with precipitation $\geq$ 1282.81 mm, dry years as those with precipitation $\leq$ 813.49 mm, and normal years as those with precipitation between 813.49 mm and 1282.81 mm (Table 3).

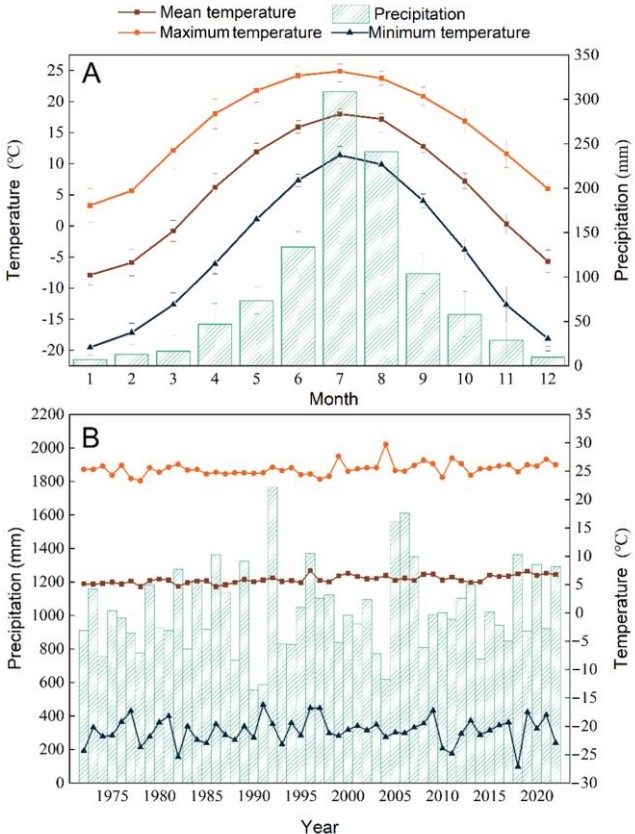

**Figure 1.** Annual changes (**A**) and interannual changes (**B**) of climatic elements in the study area.

**Table 3.** Statistical summary of annual precipitation in various hydrological periods.

| Hydrologic Period (year) | | Average Annual Precipitation (mm) |
|---|---|---|
| Wet period | 2003 to 2005 | 1506.70 |
| Dry period | 1988 to 1989 | 569.45 |
| | 2001 to 2002 | 695.75 |
| Normal water period | 1973 to 1975 | 968.57 |
| | 1977 to 1980 | 1072.88 |
| | 1982 to 1983 | 1063.75 |
| | 1991 to 1993 | 901.73 |
| | 1995 to 2000 | 1017.83 |
| | 2007 to 2011 | 1057.12 |
| | 2013 to 2015 | 935.70 |

## 2.4. Data Processing and Analysis

Statistical analyses were performed using SPSS 26.0 and Microsoft Excel 2022, with graphing completed using Origin 2022. Several methods were used to analyze the associa-

tions between PD and RP tree ring width and climate factors. The main methods included correlation analysis, multiple stepwise regression analysis, and gray correlation degree analysis. Before performing the multiple stepwise regression analysis, we performed collinearity tests on the data and determined the Variance Inflation Factors (VIF) to ensure the accuracy of the fitted equations [51]. The calculation of the gray correlation value ($R_i$) is as follows:

$$R_i = \frac{\sum L_i(N)}{n} \tag{1}$$

In this equation, $R_i$ is the gray relational degree value, $L_i(N)$ is the correlation coefficient, and $n$ is the number of reference data points.

## 3. Results

### 3.1. Statistical Analysis of Growth, Tree Ring Width and Tree Ring Chronology in PD and RP

The tree ring width of PD and RP followed the typical pattern of tree growth, initially stabilizing before decreasing (Figure 2 and Appendix A Figure A2). PD exhibited a steady growth trend until 1973, after which its growth declined. With a mean growth rate of 2.81 mm·a$^{-1}$ from 1974 to 1987, the tree ring width of PD showed a substantial decreasing trend. With a mean growth rate of 1.43 mm·a$^{-1}$ from 1988 to 2022, the growth rate of tree ring width in PD slowed. From 1972 to 1985, RP exhibited a steady growth trend in tree ring width, with a mean growth rate of 3.29 mm·a$^{-1}$. Nevertheless, since 1985, the tree ring width growth rate of RP has diminished, with a mean of 2.12 mm·a$^{-1}$.

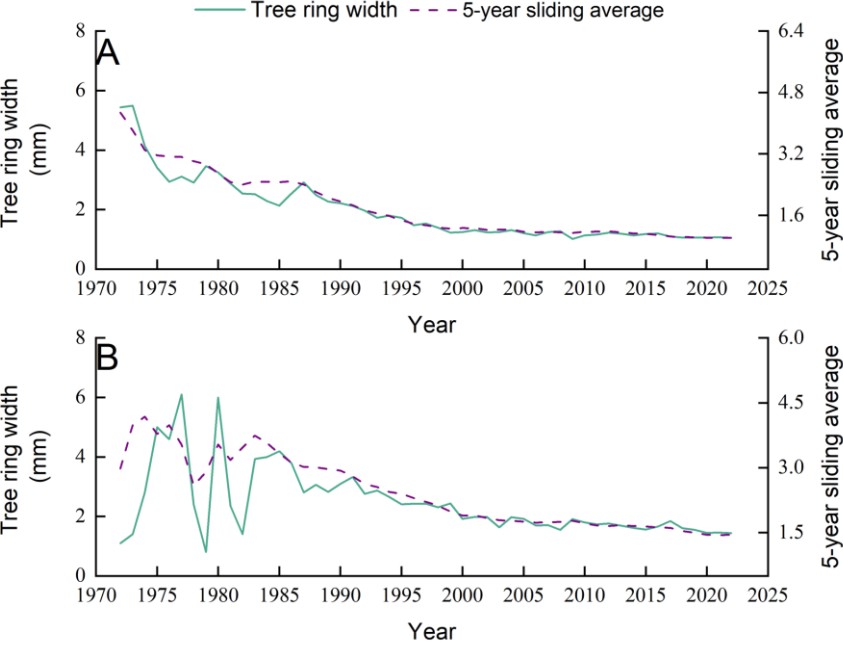

**Figure 2.** Interannual variation of tree ring width of *Pinus densiflora* Siebold & Zucc. (**A**) and *Robinia pseudoacacia* L. (**B**).

The findings indicated that the basal area increment (BAI) sequences of PD and RP observed at the sampling points exhibited different trends across various periods. The BAI sequence of PD showed a "growth-decline" change process (Figure 3A). There was no significant increase ($R^2 = 0.2236$) observed from 1972 to 1986, with the rate of increase being 15.12 mm$^2$·a$^{-1}$. However, it significantly decreased from 1986 to 2022 ($R^2 = 0.5282$, $p \leq 0.05$), with a rate of decline of 9.92 mm$^2$·a$^{-1}$. Meanwhile, the BAI sequence of RP exhibited a "growth-stabilization" change process (Figure 3B), with a highly significant increase observed between 1972 and 1991 ($R^2 = 0.7202$, $p \leq 0.01$) at a rate of 67.25 mm$^2$·a$^{-1}$. Although a downward trend was observed between 1991 and 2022, it was not significant ($R^2 = 0.0363$), with a rate of decline of 1.87 mm$^2$·a$^{-1}$.

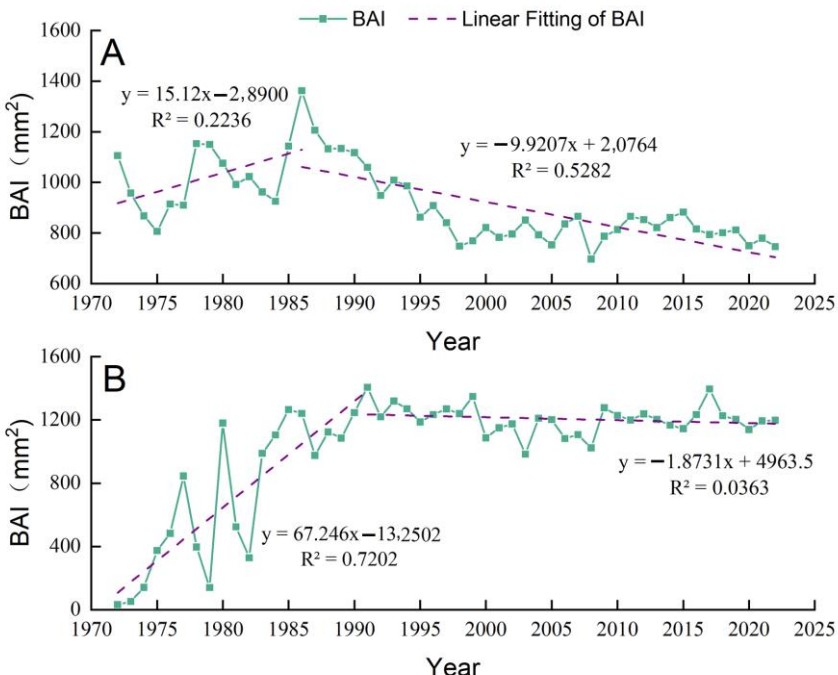

**Figure 3.** Basal area increment sequence interannual variation of *Pinus densiflora* Siebold & Zucc. (**A**) and *Robinia pseudoacacia* L. (**B**), BAI is the basal area increment.

After screening the tree ring samples of PD and RP, the basic statistical parameters of the data were analyzed. Subsequently, the STD and RES chronologies were established for both PD and RP (Table 2). The findings demonstrated that the mean sensitivity (MS), signal-to-noise ratio (SNR), and expressed population signal (EPS) in the STD were superior to those in the RES. This suggests that the STD chronology was a more reliable indicator of the response of tree ring width to climate change. Accordingly, the STD chronology was selected for analyzing the relationships between climate factors and tree ring width in PD and RP in this study. The MS surpassed the acceptable threshold (0.15), suggesting that PD and RP were highly responsive to the climate factors in the research region. The first-order autocorrelation coefficients (FOACs) of the RES chronology were higher than those of the STD chronology, and the RES chronology more accurately captured how the previous year's climate affected the tree ring width in the current year. The EPS values for all samples were above 90%, surpassing the critical threshold of 0.85. This indicates that the samples contained a significant amount of environmental information and the signals that they contained could represent the overall characteristics. The first principal component explained more than 40% of the variation, indicating that the tree growth factors were relatively concentrated. In conclusion, dendrochronological studies can be conducted in the research area using the tree ring width of PD and RP.

*3.2. Tree Ring Width of PD and RP and Single-Month Climate Factor Correlation Analysis*

The correlation analysis results (Figure 4A) revealed that the tree ring width of PD had a highly significant ($p < 0.01$) positive correlation with the $T_{max}$ in February and March of the current year; there was a significant positive correlation ($p < 0.05$) between the tree ring width of PD and the $T_{mean}$ in May of the current year, as well as the $T_{max}$ in July of the current year; and there was a significant ($p < 0.05$) positive correlation with the $P_{re}$ in December of the previous year and June of the current year. In contrast, the correlation analysis results (Figure 4B) revealed that the tree ring width of RP had a highly significant ($p < 0.01$) positive correlation with the $T_{mean}$ in October of the previous year and the $T_{max}$ in May of the current year; it was significantly positively correlated ($p < 0.05$) with the $T_{max}$ in October of the previous year and the $T_{mean}$ in July of the current year. However, the tree ring width of RP was significantly negatively correlated ($p < 0.05$) with the $T_{min}$ in

November of the previous year and the $T_{mean}$ in December of the current year. The $P_{re}$ in November of the previous year and July of the current year were significantly positively correlated with the ring width of RP ($p < 0.05$).

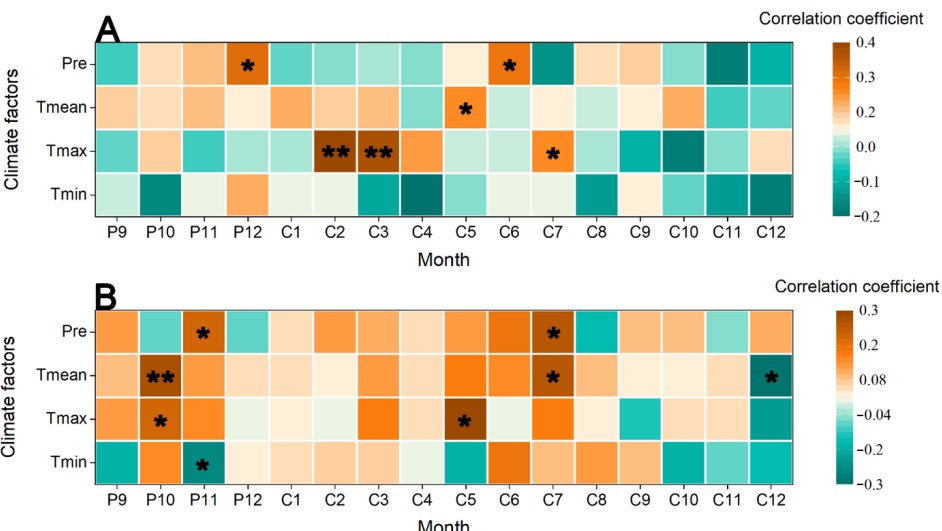

**Figure 4.** Correlation coefficients between tree ring width of *Pinus densiflora* Siebold & Zucc. (**A**) and *Robinia pseudoacacia* L. (**B**) and monthly climatic factors. $P_{re}$ is the mean precipitation per month; $T_{mean}$ is the mean temperature per month; $T_{max}$ is the mean maximum temperature per month and $T_{min}$ is the mean minimum temperature per month. *p* is for the previous year and C is for the current year. * Indicates a significant correlation at the 0.05 level and ** indicates a highly significant correlation at the 0.01 level.

### 3.3. Analysis of Tree Ring Width Response Function Results and Climate Factors for PD and RP

The variance inflation factors ($VIF_1$) corresponding to the collinearity test results of the $RWI_1$ and each climatic element are 1.71, 1.91, 1.02, 1.08, 1.89, and 1.35; The variance inflation factors ($VIF_2$) corresponding to the collinearity test results of the $RWI_2$ and each climatic element are 1.15, 1.52, 1.10, 1.02, 1.57, and 1.25. This indicates that the degree of collinearity between variables is less and can be trusted [52]. Multiple stepwise regression analyses were conducted using the identified RWI and six climate factors ($T_i$, $T_h$, $T_l$, $P_i$, $M_i$, and $W_i$) for individual years within the period 1972 to 2022. The regression equation was:

$$RWI_1 = 0.036T_i + 0.004P_i + 0.016M_i - 0.010W_i + 0.153$$
$$R = 0.675, R^2 = 45.6\%, R^2_{adj} = 40.8\%, F = 5.64, P < 0.01 \tag{2}$$

$$RWI_2 = 0.040T_i + 0.002P_i + 0.007M_i - 0.002W_i + 0.546$$
$$\left( R = 0.693, R^2 = 48.0\%, R^2_{adj} = 42.1\%, F = 4.60, P < 0.01 \right) \tag{3}$$

(In the equation, $RWI_1$ stands for the tree ring width index of PD, $RWI_2$ stands for the tree ring width index of RP, *R* is the correlation coefficient, $R^2$ is the coefficient of determination, and $R^2_{adj}$ is the correction coefficient of determination).

A closer look at this equation revealed that differences in the local climate might account for 48.0% and 45.6%, respectively, of the tree ring width in PD and RP. The obtained result demonstrated a relatively low level of explained variance compared to the mean amount of variance explained in the northern region, which was approximately 60% [53]. As shown in the contrasting trend (Figure 5), the predicted value curves for the RWI of PD (Figure 5A) and RP (Figure 5B) closely matched the actual value curves. The corresponding $R^2$ values for the two tree species were 0.70 and 0.69, respectively.

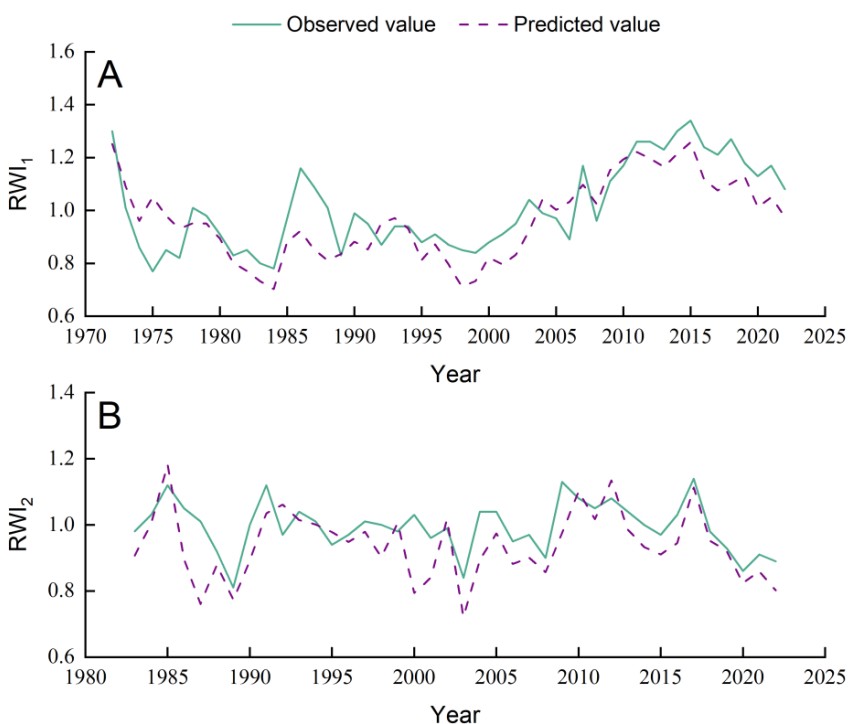

**Figure 5.** Calculated versus actual values of growth regression models for *Pinus densiflora* Siebold & Zucc. (**A**) and *Robinia pseudoacacia* L. (**B**). RWI$_1$ stands for the *Pinus densiflora* Siebold & Zucc. tree ring width index, and RWI$_2$ stands for the *Robinia pseudoacacia* L. tree ring width index.

*3.4. Analysis of the Degree of Gray Correlation between PD and RP Tree Ring Width and Climate Factors*

To address the low amount of variance explained by the response function, the four relevant climate factors, namely, $T_i$, $P_i$, $M_i$, and $W_i$, were selected for the impact comparison sequence. The tree ring width of PD and RP was chosen as the reference sequence. Gray relational analysis was performed on these four climate factors. The correlation degrees of the four related climate factors that affect the comparison sequence were obtained through gray correlation analysis (Appendix A Table A2). The following is the sequence of $R_i$ for the impact comparison series of the PD tree ring width data: $M_i$ (0.737) > $W_i$ (0.726) > $T_i$ (0.711) > $P_i$ (0.710). The following is the sequence of $R_i$ for the impact comparison series of the RP tree ring width data: $M_i$ (0.735) > $W_i$ (0.720) > $P_i$ (0.713) > $T_i$ (0.704). The results showed that temperature and precipitation were the primary climate factors affecting the tree ring width of PD and RP. These two factors considered together had a greater impact than the individual temperature or precipitation factors.

**4. Discussion**

*4.1. Temperature and Precipitation Effects on RP and PD Tree Ring Width over a Single Month*

This study of the link between the temperature in a single month and the tree ring width in various strands (Figure 4) yielded multiple key findings. The $T_{max}$ in February, March, and July of the current year, as well as the $T_{mean}$ in May of the current year, showed greater correlations with the tree ring width of PD. The $T_{max}$ in February and March had major effects on the tree ring width of PD (Figure 4A). The $T_{max}$ and $T_{mean}$ in October of the previous year, the $T_{max}$ in May of the previous year, and the $T_{mean}$ in July of the current year were all correlated with the width of tree rings in RP. The $T_{mean}$ in October of the previous year and the $T_{max}$ in May of the current year, among other variables, had significant effects on the width of the tree rings of RP (Figure 4B). This result is comparable with that of Chen, et al. [54] on the tree ring width in subtropical trees in Southeast China and their response to climate factors, as well as Zheng, et al. [55], who investigated principal



tree species' responses to climate factors in the Dabie Mountains. It is possible that trees continue to carry out photosynthesis at the end of the previous growing season, and the resulting organic compounds are stored in the trees for use during dormant periods for growth and respiration in the next year. When the temperature rises moderately in the early spring of the second year, the photosynthesis and respiration rates of the trees increase, resulting in more organic matter being stored in the trees [56]. This process triggers cell division, promotes xylem formation, and contributes to the creation of tree rings [57–59]. Temperature increases and ground thawing early in the growing season (March to May) aid in tree emergence from dormancy [60,61]. Nevertheless, the tree ring width of RP was greatly reduced by the $T_{min}$ in November of the previous year and the $T_{mean}$ in December of the current year (Figure 4B). This result is consistent with the findings of Anderson-Teixeira, et al. [62] regarding the response of tree ring width in various forest types to global climate change. One possible explanation is that the higher temperatures in winter and early spring can deplete the stored carbohydrates in broad-leaved trees, leading to growth suppression [63]. In the growing season, when trees have leaves, excessive temperature increases can stimulate photosynthesis and respiration, resulting in excessive transpiration and closure of leaf pores. These factors can negatively affect tree ring width and limit normal plant development [64].

Other analyses showed a significant positive correlation between the mean precipitation in December of the previous year and June of the current year and the tree ring width in PD. This finding is consistent with that of Pumijumnong, et al. [65] on the response of two tropical pines in Thailand to climate change and Olivar, et al. [66] on the climatology of Aleppo pine tree ring width. Most precipitation during the late growing season of the previous year is stored in the soil or runs off, providing a water source for trees in the following year. This prepares the tree for growth and is beneficial for the accumulation of more nutrients in PD, ultimately promoting increased tree ring width [67,68]. Furthermore, in the study area, tree ring width began to increase at the beginning of the growing season in April. With appropriate temperature increases and sufficient precipitation brought by the monsoon, the hydrothermal conditions required for plant growth were met, thereby increasing tree ring width [69]. However, the continuous high temperatures during the summer resulted in drought stress. The increase in precipitation during this period helped replenish the available water in the soil, ultimately promoting tree ring width [70]. Mean precipitation in November of the previous year and July of the current year were significantly and positively correlated with tree ring width in RP (Figure 4B). The photosynthesis efficiency of broad-leaved plants is reduced due to the increase in water vapor caused by the humid summer monsoon [71]. The saturation of soil with water due to excessive precipitation can reduce the soil's oxygen content and impede root uptake, leading to reduced nutrient accumulation. RP is particularly sensitive to water during growth, and excessive precipitation may create unsuitable soil conditions, decreasing tree ring width [72].

### 4.2. Interannual Variations in Temperature and Precipitation Effects on PD and RP Tree Ring Width

One of the most significant long-term environmental elements influencing the width of the rings of trees in the area is temperature change. This is because appropriate temperatures are necessary for tree life activities and to promote cell activity [73]. The RWIs of PD and RP in this area showed a general upward trend from 2000 to 2018 (Figure 6), suggesting that temperature increases have a significant impact on tree ring width in the region. The warming enhanced photosynthesis and lengthened the growing season, as reflected by tree ring width and soil nutrient availability, resulting in improved tree growth and survival rates [74]. Although a temperature rise has a favorable impact on tree ring width, this effect is constrained by the particular threshold. Continuous temperature rise can result in drought stress, which can inhibit tree ring width and cause the "temperature threshold effect" [75,76]. During the four periods of 1987–1988, 1994–1995, 2005–2006, and 2018–2019 in this region (Figure 1), there were decreases in precipitation, while the

temperature continued to increase. This phenomenon led to an extremely high degree of air dryness, which strongly limited tree ring width (Figure 6). The results align with the findings of Streit, et al. [77], who reported on the response of two conifer species at the alpine treeline. The similarity in outcomes implies that high temperatures can increase evapotranspiration from trees, reduce the available soil moisture, and have a severe impact on plant metabolic activity, thus limiting tree ring width [78].

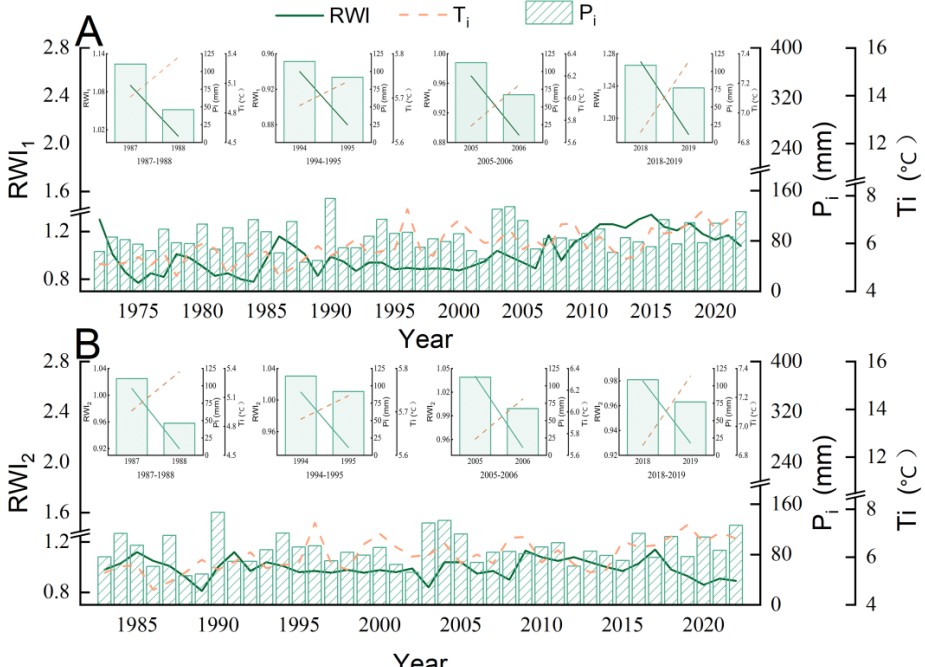

**Figure 6.** Analysis of the interannual variation trend of the tree ring width indices of *Pinus densiflora* Siebold & Zucc. (**A**) and *Robinia pseudoacacia* L. (**B**) with multiyear temperature and precipitation. $RWI_1$ stands for the *Pinus densiflora* Siebold & Zucc. tree ring width index, and $RWI_2$ stands for the *Robinia pseudoacacia* L. tree ring width index. $T_i$ is the mean yearly temperature and $P_i$ is the mean yearly precipitation.

The analysis revealed that interannual precipitation plays a crucial role in determining the RWI of PD and RP. The precipitation analysis covering the period of 1972 to 2022 shows one wet period, two dry periods, and seven normal periods (Table 3). The typical hydrological periods were 2003–2005, 1988–1989, and 1995–2000. The $RWI_1$ gradually declined during 2003–2005 (wet period) (Figure 7A), possibly due to excessive water infiltration into the soil layer, leading to root rot and impacting tree ring width [79]. Conversely, $RWI_2$ peaked in 2004 during the wet period and did not change thereafter (Figure 7B). This trend may be attributed to the ability of RP to appropriately allocate water resources by regulating stomatal openings and root water absorption capacity, thereby maintaining the water supply required for growth [80]. As time passed, when RP reached its peak water supply for growth, its tree ring width did not increase further even though the soil moisture remained sufficient. This suggests that RP can efficiently use water resources according to its growth requirements, achieving the optimal growth state without excessive consumption of water resources. The RWIs of PD and RP gradually declined during 1988–1989 (dry period) (Figure 7). This finding was comparable with that of Che, et al. [81] on the tree ring width characteristics of *Caragana korshinskii* on the western Loess Plateau, further demonstrating the general impact of drought on tree ring width. Trees require sufficient water for normal growth, and a lack of precipitation can reduce soil moisture, thereby restricting tree ring width [82]. Under extremely dry conditions, trees may adopt a "survival growth strategy" to cope with poor water conditions by reducing their growth rate to conserve water [83]. During the continuous normal period (1995–2000), the RWI

fluctuations of PD and RP were slight, suggesting that the RWI of these two tree species remained relatively stable. Moreover, the mean RWI of PD and RP during this period surpassed the annual mean RWI of the region. The reason for this phenomenon may be attributed to several factors. During the normal period, the precipitation in the area remains at a mean level, and the rate of climate change is relatively slow. This means that extreme climate factors such as prolonged droughts or continuous rainstorms are less likely to occur, and the annual mean precipitation exhibits minimal fluctuations. This climate factor keeps soil moisture at a relatively stable level. Consequently, the root systems of PD and RP can efficiently absorb adequate water and nutrients, thereby promoting their growth and development. Moreover, the growth of PD and RP is influenced by various environmental factors, which work synergistically. These factors include air temperature, light availability, and soil quality. During the normal period, these environmental factors also remain stable, further facilitating the growth and development of PD and RP [84].

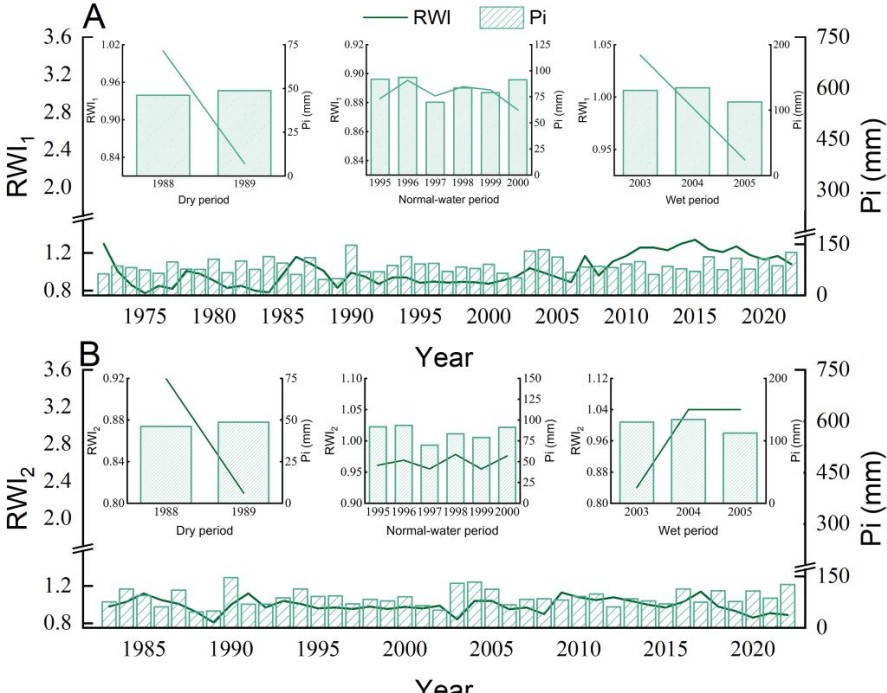

**Figure 7.** Responses of the *Pinus densiflora* Siebold & Zucc. (**A**) and *Robinia pseudoacacia* L. (**B**) tree ring width indices to multiyear precipitation under different hydrological periods. $RWI_1$ stands for the *Pinus densiflora* Siebold & Zucc. tree ring width index, and $RWI_2$ stands for the *Robinia pseudoacacia* L. tree ring width index. $P_i$ is the mean yearly precipitation.

### 4.3. Changes in the Tree Ring Width of PD and RP as a Result of Comprehensive Climate Factors

$M_i$, which is the result of the interaction between temperature and moisture, had the most significant effect on PD and RP tree ring width (Appendix A Table A2). The inferences made in this study agree with those of Liu, Gao, Sun, Niu and Wang [42] on the relationships of *Quercus acutissima* tree ring width with temperature and precipitation. The East Asian monsoon may have influenced the area, resulting in greater temperatures and abundant precipitation throughout the summer season, which created environmental conditions suitable for tree ring growth. PD and RP have shown high adaptability to the environmental conditions of hydrothermal synchronism, allowing them to quickly adjust to changes in the environment and end their dormant state promptly, thereby promoting tree growth. The warmth index represents the effective cumulative temperature in the region and can be used to assess warmth and coolness [85]. This finding further confirms the critical influence of $W_i$ on tree ring width. Previous research demonstrated that warmth-associated climate factors promote tree growth and development, photosynthesis,

and nutrient uptake, while coldness-associated climate factors hinder tree growth and development [56,86]. Additionally, $W_i$ can indirectly affect the structure of the microbial community and metabolic activities in the soil, which can impact nutrient uptake and the growth of trees [87]. Subsequent investigations showed that the tree ring width of PD displayed greater sensitivity to temperature changes than RP, which is consistent with the findings of Salomón, et al. [88]. This result may be attributed to coniferous species being better adapted to colder climates. When temperatures rise, the growth and metabolic activities of coniferous tree species may become restricted, resulting in stunted growth or other abnormalities [89]. In contrast, the tree ring width of PD exhibited less sensitivity to precipitation than that of RP (Appendix A Table A2). This difference may be attributed to the fact that PD roots typically penetrate deeper into the soil, enabling them to absorb and utilize water from the soil more efficiently. Additionally, the leaf shape and structure of PD enable it to reduce water evaporation and utilize water in the soil more effectively during drought conditions. RP generally has a larger leaf area and higher transpiration rate than PD, making the former more reliant on an adequate water supply to sustain normal growth and metabolic activities [34].

The variation trends of $M_i$ and RWI are presented in Figure 8. The change tendencies of $M_i$ and RWI exhibited good consistency (the correlation coefficients were 88.3% and 81.0%). This finding indirectly supports the precision of the grayscale analysis conducted on the relationship between the RWI and $M_i$ of PD and RP. Figure 8 also reveals a certain hysteresis effect in the impact of $M_i$ on the RWI. In the case of PD, the trends observed in $M_i$ for the periods 1975–1979 and 2006–2010 were found to be consistent with the trends observed in $RWI_1$ for the periods 1976–1980 and 2007–2011, respectively. Similarly, in RP, the trends observed in $M_i$ for the periods 1984–1987 and 2001–2004 were consistent with the trends observed in $RWI_2$ for the periods 1986–1989 and 2002–2005, respectively. This is due to a "lag effect", which occurs when changes in environmental parameters such as temperature and precipitation are not immediately reflected in the tree ring width of PD and RP [42].

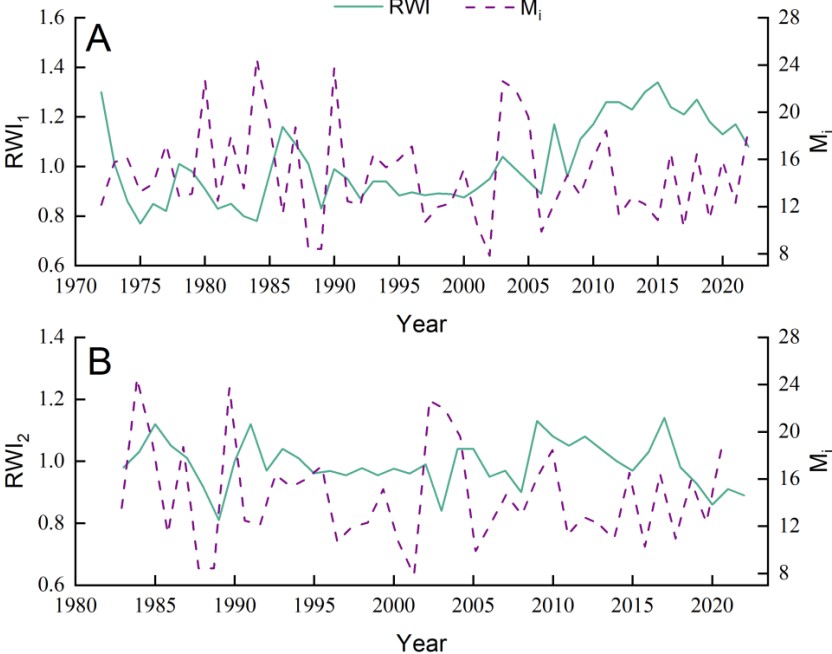

**Figure 8.** Trends in the tree ring width indices and moisture indices of *Pinus densiflora* Siebold & Zucc. (**A**) and *Robinia pseudoacacia* L. (**B**). $RWI_1$ stands for the *Pinus densiflora* Siebold & Zucc. tree ring width index, and $RWI_2$ stands for the *Robinia pseudoacacia* L. tree ring width index. $M_i$ is the moisture index that can reflect dry or wet conditions. The formula is $Mi = Pi/Ti$.

## 5. Conclusions

The respective sequences of growth rate and basal area increment for *Pinus densiflora* and *Robinia pseudoacacia* in the Mount Tai area of northern China exhibited diverse patterns of change. The study also revealed significant correlations between the tree ring chronology constructed using the tree ring width of *Pinus densiflora* and *Robinia pseudoacacia* and the climate factors in the study area. Notably, the standard chronology contained a wider range of environmental information, which is crucial for fulfilling the requirements of climate analysis and reconstruction for *Pinus densiflora* and *Robinia pseudoacacia*.

Single-month temperature and precipitation substantially affected the tree ring width of *Pinus densiflora* and *Robinia pseudoacacia*. This effect was most noticeable during the growth season. Further investigation of the associations between multiyear climate factors and tree ring width in *Pinus densiflora* and *Robinia pseudoacacia* showed that changes in hydrothermal conditions generated by global warming have a significant influence on the two tree species. Additionally, the tree ring width of these two species exhibited distinct change trends across different hydrological periods.

The moisture index was the most important climate factor influencing tree ring width, followed by the warmth index, mean yearly precipitation and mean yearly temperature.

**Author Contributions:** Conceptualization, Y.H., S.F., M.H. and P.G.; methodology, Q.Y.; software, S.F. and G.W.; formal analysis, Y.H.; investigation, Y.H. and Q.Y.; resources, D.H. and Z.L.; data curation, D.H. and Z.L.; writing—original draft preparation, Y.H.; writing—review and editing, D.H. and P.G. All authors have read and agreed to the published version of the manuscript.

**Funding:** This research was funded by Shandong Key Research and Development Program—Forest Planting Ecosystem Reconstruction and Demonstration Based on Forest Carbon Sink (grant number 2021SFGC0205), Shandong Taishan Forest Ecosystem National Positioning Observation and Research Station Operation Project (grant number 2018-LYPT-DW-053).

**Data Availability Statement:** Not applicable.

**Conflicts of Interest:** The authors declare no conflict of interest.

## Appendix A

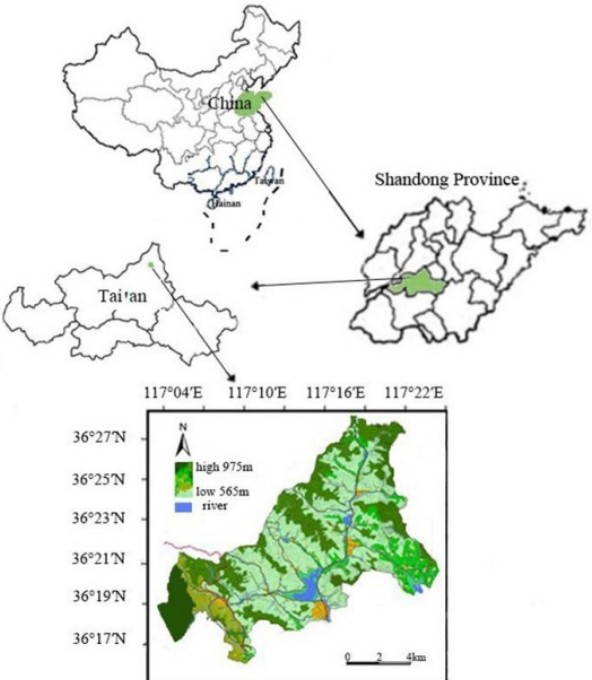

**Figure A1.** Map of the location of the study area.

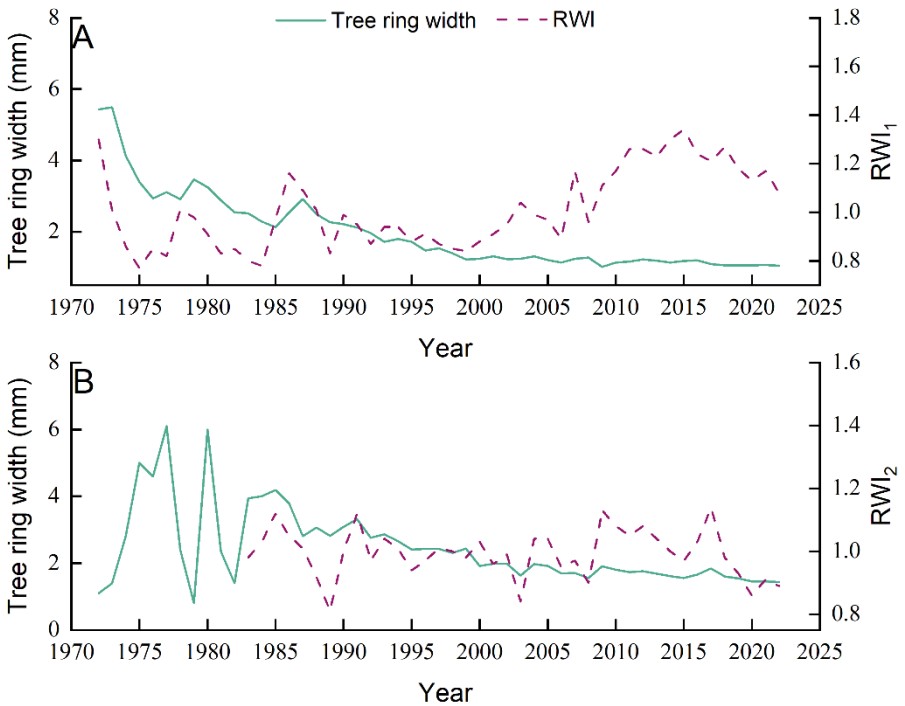

**Figure A2.** Interannual variation of the tree ring width and tree ring width indices of *Pinus densiflora* Siebold & Zucc. (**A**) and *Robinia pseudoacacia* L. (**B**). RWI$_1$ stands for the *Pinus densiflora* Siebold & Zucc. tree ring width index, and RWI$_2$ stands for the *Robinia pseudoacacia* L. tree ring width index. The sample depth for each species was 48.

**Table A1.** P-III frequency curve modulus ratio coefficient value.

| Mean Yearly Precipitation from 1972 to 2022 (mm) | P (%) | Modulus Ratio Coefficient Value | Corresponding Precipitation (mm) |
|---|---|---|---|
| | \multicolumn | $Cs = 3.0Cv$ | |
| 1042.93 | 20 | 0.78 | 813.49 |
| | 75 | 1.23 | 1282.81 |

P is the frequency corresponding to P-III, *Cs* is the runoff coefficient of the watershed, and *Cv* is the coefficient of the variation of the runoff.

**Table A2.** The value of the grey relational degree of the comparison sequence.

| Year | Pinus densiflora | | | | Robinia pseudoacacia | | | |
|---|---|---|---|---|---|---|---|---|
| | $P_i$ | $M_i$ | $W_i$ | $T_i$ | $P_i$ | $M_i$ | $W_i$ | $T_i$ |
| 1972 | 0.7605 | 0.6980 | 0.6434 | 0.6735 | 0.3371 | 0.3485 | 0.3612 | 0.3538 |
| 1973 | 0.6772 | 0.6330 | 0.7309 | 0.7216 | 0.3645 | 0.3762 | 0.3530 | 0.3548 |
| 1974 | 0.8141 | 0.9566 | 0.7973 | 0.7474 | 0.4729 | 0.5093 | 0.4682 | 0.4535 |
| 1975 | 0.4178 | 0.4273 | 0.4432 | 0.4357 | 0.5453 | 0.5587 | 0.5812 | 0.5705 |
| 1976 | 0.4282 | 0.4781 | 0.4536 | 0.4352 | 0.5835 | 0.6612 | 0.6227 | 0.5942 |
| 1977 | 0.3845 | 0.3923 | 0.3672 | 0.3556 | 0.7035 | 0.7251 | 0.6567 | 0.6263 |
| 1978 | 0.9072 | 0.8967 | 0.9345 | 0.9887 | 0.6401 | 0.6358 | 0.7141 | 0.6902 |
| 1979 | 0.6126 | 0.6042 | 0.5851 | 0.5728 | 0.5405 | 0.5460 | 0.5596 | 0.5693 |
| 1980 | 0.4094 | 0.4851 | 0.3493 | 0.3373 | 0.7144 | 0.9218 | 0.5723 | 0.5462 |
| 1981 | 0.8209 | 0.8920 | 0.9158 | 0.9361 | 0.6006 | 0.6309 | 0.6735 | 0.6488 |

**Table A2.** *Cont.*

| Year | *Pinus densiflora* | | | | *Robinia pseudoacacia* | | | |
|------|-------|-------|-------|-------|-------|-------|-------|-------|
| | $P_i$ | $M_i$ | $W_i$ | $T_i$ | $P_i$ | $M_i$ | $W_i$ | $T_i$ |
| 1982 | 0.5945 | 0.5696 | 0.6889 | 0.7036 | 0.8981 | 0.9498 | 0.7670 | 0.7525 |
| 1983 | 0.5404 | 0.5549 | 0.5862 | 0.5640 | 0.7272 | 0.7489 | 0.7963 | 0.7626 |
| 1984 | 0.7263 | 0.9734 | 0.5403 | 0.4983 | 0.8800 | 0.6713 | 0.8112 | 0.7344 |
| 1985 | 0.5789 | 0.6756 | 0.5394 | 0.4918 | 1.0126 | 0.8305 | 0.9240 | 0.8128 |
| 1986 | 0.5018 | 0.5242 | 0.5854 | 0.5675 | 0.6408 | 0.6711 | 0.7545 | 0.7300 |
| 1987 | 0.8826 | 0.8781 | 0.8422 | 0.8480 | 0.8387 | 0.8421 | 0.6773 | 0.6804 |
| 1988 | 0.5424 | 0.5575 | 0.7676 | 0.7202 | 0.5871 | 0.6018 | 0.7959 | 0.7535 |
| 1989 | 0.5921 | 0.5971 | 0.8445 | 0.8286 | 0.6394 | 0.6442 | 0.8717 | 0.8577 |
| 1990 | 0.6683 | 0.7131 | 0.7666 | 0.7968 | 0.6455 | 0.6796 | 0.9014 | 0.9359 |
| 1991 | 0.6042 | 0.6283 | 0.6726 | 0.6672 | 0.7922 | 0.8265 | 0.8902 | 0.8824 |
| 1992 | 0.7229 | 0.7434 | 0.8132 | 0.8351 | 0.8435 | 0.8665 | 0.9446 | 0.9690 |
| 1993 | 0.8435 | 0.9461 | 0.7670 | 0.7583 | 0.8936 | 0.8161 | 0.9791 | 0.9910 |
| 1994 | 0.7980 | 0.9604 | 0.8337 | 0.8526 | 0.7235 | 0.8932 | 0.7475 | 0.7599 |
| 1995 | 0.9316 | 0.8990 | 0.9639 | 0.9828 | 0.8551 | 0.8322 | 0.9448 | 0.9304 |
| 1996 | 0.9191 | 0.8401 | 0.9716 | 0.9272 | 0.7642 | 0.7178 | 0.8388 | 0.8684 |
| 1997 | 0.8261 | 0.7696 | 0.9546 | 0.9037 | 0.9851 | 0.9653 | 0.8014 | 0.7713 |
| 1998 | 0.9904 | 0.8782 | 0.8523 | 0.7980 | 0.8094 | 0.9145 | 0.7296 | 0.6961 |
| 1999 | 0.9158 | 0.8499 | 0.9718 | 0.9256 | 0.7879 | 0.8338 | 0.7201 | 0.6988 |
| 2000 | 0.7673 | 0.7806 | 0.7847 | 0.7983 | 0.7151 | 0.7246 | 0.7275 | 0.7371 |
| 2001 | 0.9326 | 0.9119 | 0.7874 | 0.8121 | 0.9407 | 0.9589 | 0.7325 | 0.7500 |
| 2002 | 0.8020 | 0.7590 | 0.8470 | 0.7525 | 0.9798 | 0.9268 | 0.7520 | 0.6886 |
| 2003 | 0.5079 | 0.4951 | 0.7681 | 0.7483 | 0.5444 | 0.5322 | 0.7776 | 0.7608 |
| 2004 | 0.5376 | 0.5558 | 0.8527 | 0.8028 | 0.5412 | 0.5564 | 0.7813 | 0.7462 |
| 2005 | 0.6288 | 0.6108 | 0.7664 | 0.8364 | 0.6039 | 0.5900 | 0.7041 | 0.7519 |
| 2006 | 0.9238 | 0.9866 | 0.6797 | 0.6608 | 0.8446 | 0.9214 | 0.6644 | 0.6494 |
| 2007 | 0.7694 | 0.8719 | 0.7177 | 0.6603 | 0.7616 | 0.8427 | 0.7193 | 0.6710 |
| 2008 | 0.7191 | 0.7049 | 0.7128 | 0.7249 | 0.7624 | 0.7492 | 0.7565 | 0.7678 |
| 2009 | 0.8523 | 0.9028 | 0.8408 | 0.7635 | 0.7135 | 0.7422 | 0.7068 | 0.6604 |
| 2010 | 0.7321 | 0.7074 | 0.7747 | 0.8043 | 0.6850 | 0.6670 | 0.7154 | 0.7361 |
| 2011 | 0.6685 | 0.6116 | 0.7887 | 0.8257 | 0.6554 | 0.6095 | 0.7477 | 0.7749 |
| 2012 | 0.9989 | 0.9574 | 0.7039 | 0.8213 | 0.9285 | 0.8817 | 0.6921 | 0.7830 |
| 2013 | 0.7511 | 0.8312 | 0.6871 | 0.6713 | 0.7376 | 0.8001 | 0.6858 | 0.6727 |
| 2014 | 0.7839 | 0.8391 | 0.6999 | 0.6760 | 0.7646 | 0.8074 | 0.6972 | 0.6775 |
| 2015 | 0.8380 | 0.9029 | 0.7092 | 0.6582 | 0.8347 | 0.8871 | 0.7262 | 0.6815 |
| 2016 | 0.5777 | 0.6595 | 0.6703 | 0.6469 | 0.5978 | 0.6687 | 0.6779 | 0.6580 |
| 2017 | 0.8924 | 0.9472 | 0.6485 | 0.6460 | 0.7724 | 0.8853 | 0.6087 | 0.6069 |
| 2018 | 0.5906 | 0.6506 | 0.6633 | 0.6570 | 0.5923 | 0.6413 | 0.6514 | 0.6464 |
| 2019 | 0.7793 | 0.8935 | 0.6477 | 0.6150 | 0.7520 | 0.8373 | 0.6472 | 0.6200 |
| 2020 | 0.5710 | 0.6391 | 0.6644 | 0.6146 | 0.5974 | 0.6579 | 0.6799 | 0.6363 |
| 2021 | 0.6853 | 0.7711 | 0.8002 | 0.6097 | 0.6979 | 0.7698 | 0.7937 | 0.6319 |
| 2022 | 0.4971 | 0.5808 | 0.6042 | 0.5885 | 0.5295 | 0.6064 | 0.6273 | 0.6133 |
| $R_i$ | 0.7102 | 0.7371 | 0.7255 | 0.7111 | 0.7133 | 0.7350 | 0.7202 | 0.7042 |

$T_i$ is the mean yearly temperature, $P_i$ is the mean yearly precipitation, $M_i$ is the moisture index, $W_i$ is the warmth index, and $R_i$ is the gray relational degree value.

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
