# Peer review of "Tree Ring Width Responses of Pinus densiflora and Robinia pseudoacacia to Climate Variation in the Mount Tai Area of Northern China"

_forests, doi:10.3390/f14102087_

Round 1

Reviewer 1 Report

Dear authors,

Thank you for this valuable and comprehensive work. It certainly extended insight for the potential and value of dendrochronological studies to assess the climate change tendencies. Such results and conclusions could be very helpful for policy and decision makers for better land-use and territorial planning.  Anyhow I would like to make a few observations, that might further improve the quality of article:

a) Statement in the lines 48-49 is not clear and should be rephrased, 

b) Figures and graphs are hardly readable, for the climatic data the network of meteorological stations or their density should be mentioned to prove the degree of their representativeness, particularly when for the mountain area, where climatic indicators are changing in short distances, 

 c) The map with the location of study area and sampling sites would essentially enrich and strength the article,

d) Did the spatial distribution of  Pinus densiflora and Robinia pseudoacacia have any transformation- vertical distribution behavioral change for 1972-2022 period or not, e.g. pine and acacia areal has shifted upward on downward due to more dry climate?

Reviewer 2 Report

The manuscript titled “Tree Ring Width Responses of Pinus densiflora and Robinia pseudoacacia to Climate Change in the Mountain Tai Area of Northern China” by Yuan He, Qinghui Yu, Guifang Wang, Ming Hao, Simin Fan, Dingmeng Hu, Zongtai Li, Peng Gao introduces dendroclimatic study of one gymnosperm and one angiosperm tree species in the North-Eastern China. Such a traditional study is mainly of interest for the regional forest management and protection, to understand climatic impact on tree growth and reaction of forest ecosystems to climate change and extremes. Authors mostly used traditional pool of methodological tools and techniques, making their approach sound. However, there are still some principal and minor issues that authors should address.

Title and abstract reflect content of the study more or less adequately; however, I would recommend usage of “climatic variation” rather than “climate change” in the title, since authors did not concentrate on long-term changes. Introduction presents relevance, state of art, and aim of the study. But in L78-85, where studies of the broad-leaved trees should be mentioned, authors returned back to conifers instead.

In the methods section, description of data is acceptable, however, addition of the map with sampling plots and meteorological data source would be appreciated, it can be put in Appendix; also, think about adding plots of raw tree ring width, ring width index, and sample depth for each year, all three combined in the same panel for one species). But methodological part should be improved a lot. First, define what option was used in ARSTAN to obtain STD chronologies (what function was used to describe and remove long-term trends). Then, warmth index is something similar to sum of active temperatures, just calculated with monthly resolution; however, did authors consider that inequality of month length can lead to biases there? In L164-169, authors briefly mentioned method that many readers are not familiar with; please provide citation and/or more detailed description, as well as reasoning for these particular thresholds (20% and 75%, I am concerned with them being not symmetrical). In L177-180, also, elaborate and cite the method, also explain meaning of N and i in Eq. 1. In L211-213 I became very confused: how RES chronology can have first-order correlation? In L237 and Eq. 2-3, listing of factors is inconsistent: seven in text, six are listed in parenthesis (and some of the abbreviations are new and unexplained here), then four included in equations. Also, the temporal interval for dendroclimatic analysis was not chosen well^ in the autumn, trees finish wood growth and cannot further react to any factors, making any further correlations (after September or October) purely random. For annual temperature and precipitation, it is better to test various intervals like August–July, September–August etc. and determine the best option (better correlating with tree ring width).

Figures and tables represent results adequately, but I strongly recommend increasing figure quality and putting all illustrative materials close to the respective piece of text instead of separate sub-section. As for textual description of the findings, it has to be gathered together properly, including some pieces pulled out of the Discussion section, especially in regards to Figures 6-8 and Table 2.

Discussion includes proper explanation and argumentation supported with relevant sources, but description of results should be severely minimized here.

Conclusions are supported by results.

Minor comments:

Latin names should be used with added author of the taxon at their very first appearances in Abstract and the main text.

L26. “This variable…” – what did you mean?

L48. “its forests” – use one or the other, but not both words.

L50. “main mode” – what did you mean?

L77–78. “such species experience unique climate factors that differ from those of coniferous species.” – both species experience the same or similar climatic conditions, it’s climatic influence that is different.

L92. “construction species” – what did you mean?

L97. Please cite mentioned “few reports”

L105 and further. “annual ring” — please use consistently term “tree ring”

L107-108. There are two chronologies, use plural form.

L109. Better use “monthly climatic factors” to show temporal resolution of series.

L111. Perhaps, word “trends” is better here than “changes”.

L120. For consistency, perhaps, use Chinese name for the Yellow River too.

L124 and further. Use “from 5,5 to 6.5” to describe the range of values.

L147. Use plural form: “are radiuses”.

L237. Use “individual years”.

L246. “contrasting trend” – what did you mean?

Table 2. Separate types of periods with horizontal lines or use aligning to the upper sided in the period type column. As of now, it is not clear what period is of what type.

The quality of English language is slipping in some places, partially because people checking it were not dendrochronologists and were not familiar with our terminology. Some of stylistic errors and places where meaning is not clear are listed in minor comments. It would be preferable if English-speaking colleague checks the text during revision (such an approach can be better than many language-correcting services, because knowledge of the field-specific terminology and common phrasing helps a lot to make meaning precise and clear).

Reviewer 3 Report

The research analyzed the tree ring width response to climate change in mountain Tai Area. The authors use multiple statistical methods to explore the tree ring width and climate factors. However, the paper has some severe problems like the followings, I will reject this paper. The top 2 are the main reasons to decline this paper.

1.      The study period is too short. The study time is from 1972 to 2022. When we ran the ASTAN, we will detrend the growing tendency. Sometimes we may need 30 years. But our whole study time is 50 years.

2.      The number of cores is not enough. For each tree species, we have only six cores (3*2 = 6). I do not think the number is reasonable to have a believable tree ring index.

3.      The regression did not exclude the collinearity. When we ran the multiple regression, the first time is to check whether there is a collinearity among the factors.

Round 2

Reviewer 3 Report

The author's reply answers all my concerns. I hope that the paper can be accepted.